# The Short-Term Effects of Mineral- and Plant-Derived Fulvic Acids on Some Selected Soil Properties: Improvement in the Growth, Yield, and Mineral Nutritional Status of Wheat (*Triticum aestivum* L.) under Soils of Contrasting Textures

**DOI:** 10.3390/plants9020205

**Published:** 2020-02-06

**Authors:** Mahendar Kumar Sootahar, Xibai Zeng, Yanan Wang, Shiming Su, Permanand Soothar, Lingyu Bai, Mukesh Kumar, Yang Zhang, Adnan Mustafa, Ning Ye

**Affiliations:** 1Institute of Environment and Sustainable Development in Agriculture, Chinese Academy of Agriculture Sciences, Beijing 100081, Chinawangyanan@caas.cn (Y.W.); bailingyu@caas.cn (L.B.); zhy2198@163.com (Y.Z.); yening_xa@cnpc.com.cn (N.Y.); 2Department of Soil Science, Sindh Agriculture University, Tando Jam 70060, Pakistan; mukeshksootar@gmail.com; 3School of Electronic and Optical Engineering, Nanjing University of Science and Technology, Nanjing 210094, China; permanand.soothar@yahoo.com; 4Key Laboratory of Crop Water Use and Regulation, Ministry of Agriculture, Farmland Irrigation Research Institute, Chinese Academy of Agricultural Sciences (CAAS), Xinxiang 453003, China; 5National Engineering Laboratory for Improving Quality of Arable Land, Institute of Agricultural Resources and Regional Planning, Chinese Academy of Agricultural Sciences, Beijing 100081, China; adnanmustafa780@gmail.com

**Keywords:** fulvic acids, soil fertility, soil organic carbon, organic–inorganic composite, plant growth, available nutrients

## Abstract

Fulvic acids (FAs) improve the structure and fertility of soils with varying textures and also play a crucial role in increasing crop production. The pot experiment was carried out using wheat grown on three soils with a silty clay, sandy loam, and clay loam texture, respectively. The soils were treated with FAs derived from plant and mineral materials. Plant-derived solid (PSFA), mineral-derived liquid (NLFA), and plant-derived liquid (PLFA) were applied at a rate of 2.5, 5, and 5 g kg^−1^ and control applied at 0 g kg^−1^. The results showed that in treated soils, the heavy fraction C was higher by 10%–60%, and the light fraction C increased by 30%–60%. Similarly, the available N content significantly increased in treated soils by 30%–70% and the available K content increased by 20%–45%, while P content significantly increased by 80%–90% in Aridisols and Vertisols and decreased by 60%–70% in Mollisols. In contrast, for P, the organic–inorganic compounds were greater in Aridisols and Vertisols and lower in Mollisols. However, organic–inorganic composites decreased in Vertisols relative to the other two soils. Further results showed that PSFA and NLFA accelerated the plant growth parameters in Mollisols and Aridisols, respectively. Our study demonstrates that the application of PSFA and NLFA had a positive effect on the physical and chemical properties and plant growth characteristics of Mollisol and Vertisol soils. Moreover, the application of solid-state FA yields better results in Mollisols. However, liquid FA increases the nutrient availability and the effects on the chemical, biological, and physical properties of Aridisol and Vertisol soils.

## 1. Introduction

The structures and inherent fertility of soils are important features that govern sustainable crop production, particularly under intensively managed agroecosystems [1]. It is well established that the texture of soil containing varying types of clay minerals affects crop production. Vertisols are characterized by high swelling and shrinkage capacity soils that stop crop production through alternate wetting and drying periods [2]. This poor soil structure restricts the application of Vertisols to sustainable agricultural practices and makes management difficult for engineering purposes as well [2,3]. By virtue of their parent material, Vertisols are rich in calcium carbonates, which causes a calcareousness of soils that prevents the uptake of essential nutrients by influencing the physico–chemical and biological properties of the soils [4]. Despite these limitations, these types of soils are broadly exploited all over the world for crop production. Locally, Vertisols soils are called Shahjiang black and cultivated around 4 × 10^−6^ near the Huai River in the northeast part of China [2]. Mollisols accumulate illite and vermiculite minerals, organic–inorganic complexes, and demonstrate an increasing cation exchange capacity (CEC), leading to the replication of redox conditions in Mollisols [5,6] that would otherwise regulate key physical, chemical, and biological processes. Improper land management, soil acidification resulting from extensive fertilization, and low microbial biodiversity have contributed to a considerable reduction in the soil productivity of Mollisols [7,8]. Mollisols (called Albic soils) in China are distributed primarily in the northeast region, which is one of the three largest areas in the world in which Mollisols occur naturally [9]. Due to elevated soil aggregate destruction, wind and water erosion are associated with the loss of clay particles and subsequent deposition of silt/sand. These soils exhibit poor nutrition for supporting crop production [10]. Although such soils are known for their high inherent fertility [11], a significant decline in soil productivity has been observed due to inappropriate agricultural practices, such as irrigation and fertilization [12]. Therefore, it is necessary to find alternate strategies in order to improve the productive capacity of such soils to realize sustainable agricultural production.

Aridisols are typical cultivated soils of arid and semiarid regions and are distributed around the world. The Hexi Corridor and Yellow River basin are the most representative area of Aridisols in China and occupy half of their total area [13]. Although Aridisols are considered the most productive soils, they are also the most nutrient-demanding soils in China [13]. Water availability, salinization, and wind erosion threaten the productive capacity of Aridisols in agriculture [14,15]. They are also vulnerable to erosion, which depletes the top soil, thereby lowering the content of clay and organic matter and the availability of macro and micronutrients [16,17].

A significant number of studies have focused on the application of manure (mainly of animal origin), crop straw return, and mineral fertilization in the above-referenced soils to sustain soil and crop productivity. A hitherto unexploited area within this context is the application of humic substances, which have shown promise to improve the productivity of soils and the establishment of crops under soils of varying mineralogical compositions [5,18,19]. Humic substances (HSs) comprise ~60% of soil organic compounds; these compounds are considered to be a crucial constituent of the agro-ecosystem and are responsible for numerous composite and chemical reactions in the soil [20]. They are rarely decayed because of their interactions with inorganic segments [21]. In soil, one of the most prominent characteristics of HSs is their ability to interact with metallic ions, oxides, hydroxides, minerals, and carbon-based compounds [22]. Moreover, HSs can interact with xenobiotic organic molecules, such as pesticides, and affect micro and macronutrients [23]. Humic substances become more effective when they are separated into their constituent forms, e.g., fulvic (FA), humic, and humin acids. Humic substances can enhance various soil properties, such as the water holding capacity, through increased aggregation and enhanced soil aeration, leading to increased microbial activity that, in turn, increases the mineralization of organic matter and enhances the bioavailable fractions of both micro and macronutrients for plant uptake [5,24]. Various researchers have revealed that such humic substances enhance root, leaf, and shoot growth by stimulating the germination of various crop species [25]. The application of FA increases nutrient uptake [25,26] and cell permeability [27] and appears to regulate key plant metabolic mechanisms [28].

Importantly, the addition of organic additions to improve soil properties and enhance crop growth and productivity are governed by the quantity, type, and properties of added humic substances. Furthermore, mineralogical and soil textural differences may vary the effects on plant growth [29]. We hypothesized that application of various FAs (mineral- and plant-derived solids/liquids) would cause different improvements to plants’ growth and soil properties in soils with contrasting textures (i.e., Mollisols, Aridisols, and Vertisols). The objectives of our study were to measure the effects of mineral- and plant-derived FAs on the selected soil properties and to determine their effects on the development of the growth parameters, yield, and mineral content of wheat grown on Mollisols, Aridisols, and Vertisols. We examined the interaction of fulvic acids and soil types, as these plant and mineral types of fulvic acids have been the least studied to date.

## 2. Results

### 2.1. Effect of FA on Electrical Conductivity and Soil pH

The effects of the FAs on electrical conductivity and soil pH are presented in Figure 1. The electrical conductivity increased between the treatments and decreased with the NLFA in Mollisols compared to the control. The electrical conductivity increased by 10.5% and 0.18% and decreased by 14.1% in Mollisols but increased by 96.8%, 130.5%, and 195.2% in Aridisols and by 57.5%, 10.1%, and 104.7% in Vertisols between the applied treatments and the control (Figure 1A). However, soil pH decreased in Mollisols and Aridisols and increased with NLFA in Vertisols between the treatments and the control. The soil pH decreased by 5.5%, 12%, and 12.6% in Mollisols; 3.59%, 2.35%, and 3.03% in Aridisols; and 1.4% and 1.53% in Vertisols; and increased by 0.25% with NLFA in Vertisols compared to the control (Figure 1B).

### 2.2. Effect of FAs on Heavy and Light Fractions of Organic Carbon

The light and heavy fractions of labile organic carbon after the application of FAs are presented in Figure 2. The heavy fraction carbon decreased in Mollisols and with PSFA in Vertisols and increased in Aridisols and Vertisols in the treatments compared to the control. The heavy fraction carbon decreased by 36.8%, 17.9%, and 19.6% in Mollisols, increased by 51.3%, 59.5%, and 53.4% in Aridisols and 5.14% and 10.6% in Vertisols, and decreased by 3.54% in Vertisols between the treatments and the control (Figure 2A). However, the light fraction carbon increased in Mollisols and Vertisols and decreased in Aridisols relative to the control. The light fraction carbon content increased by 69.6%, 94.7%, and 12.8% in Mollisols and by 36.4%, 76.8%, and 46.3% in Vertisols and decreased by 17.6%, 12.9%, and 12.3% in Aridisols between the treatments and the control (Figure 2B).

### 2.3. Effect of FAs on Soil Chemical Properties

The influences of the FAs on the chemical properties of the soil are presented in Figure 3. The soil organic carbon content increased in Mollisols and Vertisols and decreased in Aridisols with PSFA in the treatments compared to the control. The soil organic carbon content increased by 16%, 37.8%, and 5.7% in Mollisols; by 1.68% and 24.3% in Aridisols; and by 8.7%, 32.5%, and 10.4% in Vertisols; and decreased by 11.7% with PLFA in Aridisols between the treatments and the control (Figure 3A). However, the available nitrogen (AN) content increased in between the treatments and the control within the soil. The AN content increased by 32.2%, 25.4%, and 63.6% in Mollisols; 25.2%, 76%, and 76.1% in Aridisols; and 0.57%, 4.83%, and 10% in Vertisols between the treatments and the control (Figure 3B). Similarly, the available phosphorus (AP) content decreased in Mollisols and increased in Aridisols and Vertisols. AP content decreased by 31.1%, 70.3%, and 62.2% in Mollisols and increased by 206.1%, 207.2%, and 93.2% and 13.6%, 185.6%, and 25.3% in Aridisols and Vertisols, respectively, compared to the control (Figure 3C). The available potassium (AK) content increased in the treatments compared to the control in the soil. AK content increased by 44.9%, 32.9%, and 43.3%; 23.4%, 12%, and 30%; and 3.53%, 2%, and 9.5% in Mollisols, Aridisols, and Vertisols, respectively, in the treatments compared to the control (Figure 3D).

### 2.4. Effect of FAs on the Organic–Inorganic Compounds and Their Complexes

The effects of the FAs on organic–inorganic degree compounds and organic–inorganic composites are presented in Table 1. The organic–inorganic degree compounds increased in Aridisols and Vertisols; however, they decreased in Mollisols in the treatments compared to the control. The organic–inorganic degree compounds increased by 30.7%, 31.5%, and 46.6% in Aridisols and 23%, 40.7%, and 29% in Vertisols and decreased by 37.7%, 31%, and 13.7% in Mollisols in the treatments compared to the control. However, the organic–inorganic compound content increased in Aridisols and in Mollisols with PLFA. The organic–inorganic compound content increased by 13.3% with PLFA in Mollisols and by 57.1%, 57%, and 50% in Aridisols and decreased by 26.6% and 6.66% in Mollisols and by 13%, 15%, and 9% in Vertisols in the treatments compared to the control (Table 1).

### 2.5. Effect of FA on Wheat Grain and Spike Grain weight (g pot^−1^)

The effects of FAs on the grain weight and spike grain weight of wheat are given in Table 2. The wheat grain weight decreased in Aridisols and Vertisols and increased with PSFA in Mollisols in the treatments compared to the control. The wheat grain weight increased by 1.60% with PSFA and decreased by 6.43% and 5.03% with NLFA and PLFA in Mollisols; however, the weight decreased by 15.6%, 8.25%, and 15.2% in Aridisols and by 11.4%, 13.4%, and 6.17% in Vertisols between the treatments and the control. The spike grain weight increased with PSFA and PLFA in Mollisols and with NLFA in Aridisols and decreased in Vertisols in the treatments compared to the control. The spike grain weight increased by 23.6% and 10.8% in Mollisols and by 13.5% in Aridisols and decreased by 2.40% in Mollisols, 60.4% and 10.6% in Vertisols, and 6.19%, 4.22%, and 42% in Vertisols in the treatments compared to the control (Table 2).

### 2.6. Effect of FAs on Plant Growth and Biomass Accumulation

The plant height and plant biomass content of wheat after the application of FA are given in Table 3. The plant height increased in Mollisols and Aridisols with PSFA and NLFA and decreased in Vertisols in the treatments compared to the control. The plant height increased by 11.2%, 7.67%, and 4.34% in Mollisols and increased by 3.33%, 28.7%, and 7.83% in Aridisols and by 13.5%, 98.4%, and 7.82% in Vertisols in the treatments compared to the control. The plant biomass content increased in Mollisols, Aridisols, and Vertisols and decreased with PSFA in Aridisols and Vertisols compared with the control. The plant biomass content increased by 21%, 11.8%, and 28.4% in Mollisols; by 8.48% and 5% in Aridisols; and by 22% and 1.44% in Vertisols and decreased by 46.1% and 14% with PSFA in Aridisols and Vertisols compared with the control (Table 3).

### 2.7. Effect of FAs on the Nutrient Content of Wheat

The uptake of wheat nutrients after the application of FA to Mollisols, Aridisols, and Vertisols is presented in Figure 4. The total nitrogen (TN) content increased with PSFA and NLFA in Mollisols and with PSFA in Aridisols but decreased in Vertisols in the treatments compared to the control. The TN content increased by 36.6% and 9.1% in Mollisols and by 5.93% in Aridisols and decreased by 36.4%, 47.3%, and 32.4% in Vertisols in the treatments compared to the control (Figure 4A). However, the total phosphorus (TP) content increased with NLFA and PLFA in Mollisols and with PSFA in Aridisols but decreased in Vertisols in the treatments. The TP content increased by 5.84% and 10.7% in Mollisols and by 8.23% with PSFA in Aridisols and decreased by 2.3% and 6.17%, 6.79% and 21.7%, and 63.77% and 11.7% in Mollisols, Aridisols, and Vertisols, respectively, in the treatments compared to the control (Figure 4B). The total potassium (TK) content significantly increased in the treatments of Mollisols and Vertisols and non-significantly decreased in Vertisols. The TK increased by 8.98% and 7.86% in Mollisols, by 36.1% and 3.07% in Aridisols, and by 30% and 0.73% in Vertisols. The TK similarly decreased by 9.36% with PSFA in Mollisols and by 8.05% and 1.46% in Vertisols between the treatments and the control (Figure 4C).

## 3. Discussion

### 3.1. Influences of FAs on the Physiochemical Properties of Mollisol, Ardisol, and Vertisol Soils

Soil electrical conductivity and pH are important soil quality parameters. Soil pH and electrical conductivity directly affect the buffering capacity of soils, regulate soil acidification, and release carbon dioxide [30]. Because acidic FAs remain in the soil solution, FAs are soluble in water irrespective of the pH conditions [31]. Our study demonstrated that FAs increased the physiochemical properties of all soil types studied in the current experiment. It was observed that the electrical conductivity significantly increased in Mollisols, Aridisols, and Vertisols irrespective of the applied treatments, except for the NLFA in Mollislos (Figure 1A). Similarly, FAs significantly decreased the pH of Mollisol and Ardisol soils and increased the pH of Vertisols with NLFA (Figure 1B). The increases in electrical conductivity between the treatments among the soils are related to the dominance of ion dissolution over biological assimilation or ion pair formation [32]. However, the variability in pH values can be attributed to the transformation of the substrate from a reduced to an oxidized material during mineralization and to the intense liberation of CO_2_ [33]. The predominance of alkaline (compared to mineral) pH suggests that HS established a buffer system, which is a characteristic of FA molecules [34]. Inverse results were obtained by Santio et al. [35], who reported that a decrease in electrical conductivity and an increase in soil pH values were observed during the mineralization of humic substances (HA/FA) during incubation. FAs are organically produced substances that release soil nutrients and make them available for plant use [36]. Similarly, Horie et al. [37] reported that the treatments of organic alterations enhanced the soil’s physical structure and also improved its nutrient content. Our results indicate that FA application can significantly improve the soil organic carbon content of Mollisols, Aridisols, and Vertisols (Figure 3A). The increases in soil organic carbon content among various soils with different textures caused by the application of FAs may be due to the fact that FAs are organic compounds that directly enhance the soil organic matter content in Mollisols and Vertisols and, therefore, increased organic matter mineralization [24]. Similarly, FAs significantly increased the N content of Mollisols, Aridisols, and Vertisols. Nitrogen was highly observed in the PLFA of the soils (Figure 3B). These results are similar to the results of Boguta et al. [38], who reported that the soil application of humus had a significant effect on N content. Similarly, Defline et al. [39] investigated the influence of humic acid on the form of a foliar spray along with N content increases and a greater uptake of N in corn. Likewise, available P significantly increased in Aridisols and Vertisols and decreased in Mollisols between the treatments and the control (Figure 3C). The solubility of the phosphorus concentration in the soil can be increased by the FAs. For instance, if an aluminum molecule is bound on phosphorus, FAs break the bond between these molecules, making the phosphorus available for the plants [34]. However, a decrease in Mollisols could occur because organic products in these types of soils reduce P absorption in competition with phosphorus adhering to the surface of humic acid, which behaves as a chelating agent, such as ethylenediamine, o-Hydroxyphenylecetic (EDDHA) [38,40]. These results agree with those of Wang et al. [41], who reported that the accumulation of HA in soil with P fertilizer significantly increases the amount of water-soluble phosphate, steadily retarding the foundation of occluded phosphate and improving P uptake and yield. However, Cimrin and Yilmaz [42] observed that the application of humic acid did not increase the P content a lettuce plant. On the other hand, the available K content was significantly increased in the soil in the treatment compared to the control conditions. A higher K content was detected with PLFA (Figure 3D). The increase of the K content among the soils could be because FAs release the fixed K and make it available for plants. Moreover, FAs may expand the clayey soil and release K [41]. These results are in line with those of Imbufe et al. [43], who observed that the application of humic acid and K fertilizer significantly increased the K content with 100 ppm of humic acid and 300 kg/ha K fertilizer.

### 3.2. The Influence of FAs on Heavy and Light Fraction C and Organic Complexes

The soil organic carbon dynamic is generally defined by separating soil organic matter (SOM) into two or more fractions. The SOM’s physical fractionation features distinctive specific C pools responsive to managing and classifying the physical control of SOM [44]. Previously, it was observed that labile pools of soil organic matter are more complex than total soil organic C pools due to the cropping practices in temperate soils [45]. The light fraction is generally considered plant-like, with a minimal quantity exhibiting a high C concentration [46], a light fraction sensitive to change, and strong management practices [46]. Nevertheless, the light fraction also correlates well with the rate of N mineralization [47]. However, heavy fraction C has low C concentrations and is more stable at a high density [48,49]. Heavy fraction C can be a major problem for soil fertility because it contains mineralizable C [50]. In our study, heavy fraction C increased by 50% to 60% in Aridisols, 5% to 10% in Vertisols, and decreased by 20% to 36% in Mollisols in the treatments compared to the control. A higher amount of the heavy fraction was observed in the PLFA compared to the control (Figure 2A). Similarly, the light fraction C content significantly increased by 60% to 90% and 40% to 70% in Mollisols and Vertisols, respectively, and decreased by 12% to 15% in Aridisols (Figure 2B). The increases and decreases of heavy and light fraction C among the treatments could be because FA enhanced the soil organic matter content. Organic matter also improved the organic matter mineralization within the soil. We observed that SOM was solubilized by soil organic materials when the sample was processed through NaI during the sonication and densitometric analyses, which could be the reason for the decreases and increases in light and heavy fraction C between the soils [51]. These conclusions are similar to those of Crow et al. [52], who reported that the recombined density fraction C was observed to be 40% in bulk soil compared with light fraction C during a one-year incubation experiment. Crow et al. [52] also found that heavy fraction C decomposed more than light fraction C during the experimental periods. Similarly, vo Lutzow et al. [53] observed that precipitation treatments stimulated the light fraction C, the ratio of the light/heavy fraction, and artificial warming, which reduced soil labile C during the soil labile and recalcitrant C experiments.

The organic–inorganic composite is the structural basis of soil fertility, and the organic–inorganic degree complexes are an important indicator for evaluating soil fertility [54]. In our study, the organic–inorganic degree compound content significantly increased under the treatments of Aridisols and Vertisols and decreased in Mollisols. Similarly, the organic–inorganic composite presented significantly lower amounts among the soils in the treatments compared to those of the control (Table 1). The increases and decreases in organic–inorganic complexes and organic–inorganic composites may have been caused by the different sources of FAs and the different FA concentrations applied [55].

It has been reported by many scientists that FA increases nutrient content, enhances plant growth, reduces the uptake of perilous elements, and improves plant metabolism [56]. However, to date, the effect of FA in different forms on soil organic fractions, available nutrients, plant growth, and nutrient uptake has not been studied for Mollisol, Aridisol, and Vertisol soils. The application of lignite-derived HA increased plant growth, grain weight (19% to 58%), and NPK uptake in plants [57]. The present study indicates that the application of NLFA and PLFA FA increases plant growth and nutrient content in soils. These results support the findings of Arjumend et al. [57], which show that lignite-derived HA significantly increases the plant growth and nutrient content of wheat in loam and silty loam soils.

### 3.3. Influence of FAs on the Plant Growth Characteristics of Wheat

The increase in plant growth, plant biomass, thousand-grain weight, and spike grain weight under NLFA and PLFA FA treatments between the soils confirms that FA directly influences plant growth characteristics (Table 2 and Table 3). The application of PSFA significantly increases the plant growth, plant biomass, and thousand-grain weight content in Mollisols compared to the other two soils. However, NLFA and PLFA performed better in Aridisols and Vertisols. The increase in growth parameters due to PSFA, NLFA, and PLFA may be caused by the application of these FAs, which are completely dissolved in soils and directly absorbed by the roots. Another reason could be that the application of liquid FA directly affects the metabolic sites and plant cells. These liquid FAs polarize the soil and release the nutrients of plants because of soil microbes [58]. These results are in good agreement with those of Defline et al. [39], who observed that HA significantly increases the growth and yield of corn. However, other researchers have observed that the foliar- or soil-based application of FA/HA temporarily increased plant production and plant dry mass compared to the control [59]. Our results revealed that PSFA, NLFA, and PLFA significantly increased the NPK uptake of wheat in Mollisols and Aridisols. However, Vertisols were decreased compared to the control (Figure 4). The decreases of the nutrient uptake in Vertisols could be due to the compaction of soils, which prevents the nutrients in soils from penetrating into the roots. Similar results were obtained by Arjumend et al. [57], who reported that the application of HA to soil significantly increased aspects of plant growth, such as the shoot length, shoot dry weight, and plant height, compared to the foliar-applied HA. Similarly, Eshwar et al. [60] observed that the foliar application of FA along with FeSO_4_ increased the nutrient uptake and dry matter yield of aerobic rice. The results of this study show that the application of liquid forms of FA increase the plant growth and soil properties of Aridisols and Vertisols. However, PSFA had a strong impact on Mollisols. This result supports the conclusions of Khaled et al. [61], who observed that HA significantly increased cell permeability, showing that the plant membrane can improve root development in sandy soils. Similarly, Celik et al. [62] concluded that HA can inhibit plant growth and the uptake of nutrient elements; HA also ameliorated damaged soil properties and, when combined with fertilizers, increased agricultural yield and plant growth.

## 4. Materials and Methods

### 4.1. Collection and Preparation of Soil and Plant Samples

The soils used in the present experiment were collected from three different fields/locations situated in the Anhui, Gansu, and Heilongjiang provinces in China. These soils were characterized as having typical silty clay, sandy loam, and clay loam textures. The soils were previously cultivated with summer maize and winter wheat crops with an area of approximately 580 farms. The soils were taken at a depth of 0–20 cm in the topsoil of agricultural lands. To determine their chemical and physical properties, the soils were air-dried, ground, and sieved with a 5-mm sieve to eliminate plant debris and rocks. Their physical and chemical properties were then analyzed. The soil texture was determined by the hydrometer method [63]. Electrical conductivity (EC) and pH were analyzed in the soil water extract (1:2.5 *w*/*v*) [63]. Organic matter content was analyzed by a modified Walkley–Black method [64]. Available N was determined by the Kjheldal method using a Buchi K-437/K-350 digestion/distillation unit [63]. Available P was determined using a Shimadzu UV 1208 model spectrophotometer according to the Olsen method [63]. Exchangeable cations (Ca, K, and Mg) were extracted using ammonium acetate at pH 7.0 [32] and were determined using an Eppendorf Elex 636 model flame photometer. Five to 10 flag leaves of wheat were selected for their N, P, and K content. The leaf tissues were ground and passed through a 0.2-mm sieve, digested with the mixture of acids (H_2_SO_4_ + HCLO_4_) [65], and diluted with DI water. The samples for P and K were run through an ICP Mass spectrophotometer (ICP-OES) (Thermo Fisher Scientific, 7000 Series, Waltham, MA, USA). The chemical and physical properties of the soil are presented in Table 4. All the chemicals used were of analytical grade and provided by Master Liao Company, Beijing, China. The soil samples were collected from the Heilongjiang, Gansu, and Anhui provinces of the country (Figure 5).

### 4.2. Organic–Inorganic Compound and Organic–Inorganic Composite Analysis

The soil labile and recalcitrant fractions of soil organic matter were separated by density fractionation, which is one of the physical fractionation methods widely used across the world. The light fraction C of low density (<1.7 g cm^−3^) was partly decayed plant and animal products, and the heavy fraction of a high density (>1.7 g cm^−3^) consisted of mineral-associated humic substances [34]. Approximately 5 g of the sample was placed in a centrifuged tube with 25 mL of NaI solution at a density of around 1.7 g cm^−3^. Samples were shaken in a mechanical shaker for 30 min. Then, the tubes were transferred to a centrifuge at 3000× *g* rpm for 10 min. A fiber glass filter in a Buchner funnel was used to separate the light fraction that is typically present on the surfaces of soils. This process was repeated 2–3 times to separate the light and heavy fractions. Residual material was washed with distilled water 3–4 times on a centrifuge machine. After rinsing with water, the light fractions were washed with 0.01 M CaCl_2_ and then 25 mL of deionized water. The light and heavy fractions were dried at 60 °C for 48 h, weighed, and ground to determine their C content, after which, the fractions of both organic and inorganic complexes were calculated by the following equations:Organic–inorganic compound% = HC × HW/SW × SC × 100,(1)
Organic–inorganic composite g/kg = HC × HW/SW,(2)
where HC (g/kg) is the carbon content in the heavy fraction, HW (g/kg) is the tube weight subtracting the final weight of soil after the removal of the light and heavy fraction, SW (g) is the weight of the air-dried soil, and soil organic carbon (g/kg) is the content of soil organic carbon.

### 4.3. Experimental Design and Crop Maintenance

A pot experiment was carried out on the Experimental Farm of the Institute of Environment and Sustainable Development in Agriculture (IDEA, CAAS), located in Shunyi, Beijing, China (40°09′ N, 116°92′ E). Pots of 30 × 50 cm in size were filled with air-dried soil, and 15 seeds of wheat were sown separately in the homogenized soil (15 kg). The pots were arranged in light humid conditions under a randomized complete block design (RCBD) for the entire growing season (from November 2017 to June 2018). To maintain the field capacity of the soils, the pots were watered weekly from November to January 2018 and simultaneously from February to June. The compound fertilizer (high tower) was applied at a rate of 15 g pot^−1^ with a composition of 30% N, 9% P_2_O_5_, and 9% K_2_O.

### 4.4. Collection of Plant- and Mineral-Derived FA

The experiment consisted of fulvic acids (FAs) extracted from both plant (vegetables, flowers, and fruits) and mineral sources, which are the by-products of aluminum sulfite produced by (Shandong Quan Linjia fertilizer Co. Ltd., Linyi, China) with a purity of 300 g/L in which N + P_2_O_5_ + K_2_O ≥ 80 g/L. The present study included four concentrations of FA 0 for the control and 2.5, 5, and 5 kg/ha of plant-derived solid (PSFA), mineral-derived liquid (NLFA), and plant-derived liquid (PLFA) FA, respectively, with three replications. The initial material of the FAs was derived by the International Humic Acid Society (IHSS) method [19]. More details on fulvic acid application can be found from [18].

### 4.5. Elemental Analysis of FAs

The elemental makeup of FAs like nitrogen (N), carbon (C), hydrogen (H), and sulfur (S) (CHNS) was determined by combustion in oxygen gas using a Parkin-Elmer 240-C elemental analyzer. The elemental ratio was determined among different elements, where N, C, H, and S are the atomic ratios, as reported by Kumada (1987), using Equation (3): The elemental composition of plant and mineral-derived fulvic acids are presented in (Table 5).
O% = 100 − (N% + C% + S%).(3)

### 4.6. Calculation and Statistical Analyses

Statistical tests were performed using SPSS (SPSS Version 21.0, Chicago, IL, USA). A one-way analysis of variance (ANOVA) was performed on the soil and plant parameters (SOC, N, P, and K, spike length, plant height, and biomass). Data are expressed as the means ± SE (standard error). The means (*n* = 4) were subjected to a one-way analysis of variance (ANOVA) and compared using Bonferroni multiple comparison tests at a significance level of *p* < 0.05. The location of the experimental site is shown by a map drawn in Corel Draw m version X7.

## 5. Conclusions

The present study demonstrated that the application of FAs improved the physico-chemical properties of soils with varying texture. In addition, the application of FAs increased the nutrient uptake and production of wheat grown on these soils. Along with the soil organic carbon fractions, FAs also had a great impact on the growth characteristics of wheat. Overall, we determined that PSFA exerted a large effect on the properties of Mollisols. On the other hand, NLFA and PLFA increased the properties of Aridisol and Vertisol soils. It is therefore recommended that the application of mineral- and plant-derived FAs should be studied at the field level, and the underlying mechanisms that exert variable effects on the soil properties should be explored in detail.

## Figures and Tables

**Figure 1 plants-09-00205-f001:**
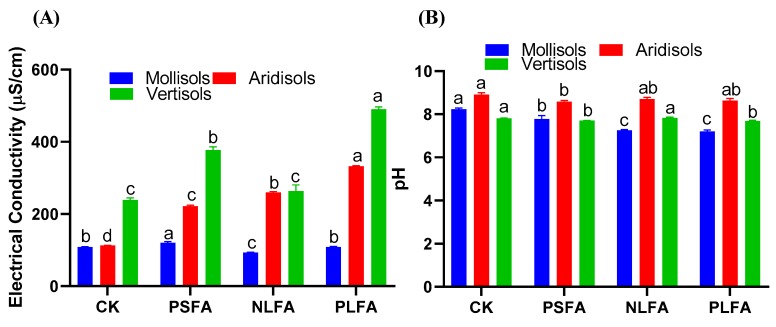
The effect of plant-derived solid fulvic acid (PSFA), mineral-derived liquid fulvic acid (NLFA), and plant-derived liquid fulvic acid (PLFA) on (**A**) electrical conductivity and (**B**) soil pH of Mollisol, Aridisol, and Vertisol soils. The means ± standard errors are shown (*n* = 4). Different letters (a, b, and c) indicate significant differences between the initial and final values of the soils based on a least significant difference test at a 5% significance level.

**Figure 2 plants-09-00205-f002:**
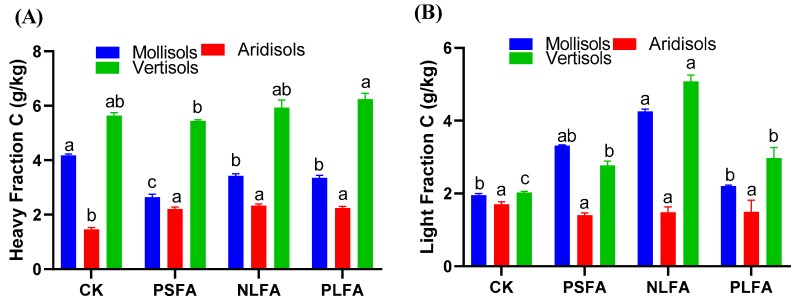
The effect of plant-derived solid fulvic acid (PSFA), mineral-derived liquid fulvic acid (NLFA), and plant-derived liquid fulvic acid (PLFA) on (**A**) heavy fraction C and (**B**) light fraction C content of Mollisols, Aridisols, and Vertisols soils. The means ± standard errors are shown (*n* = 4). Different letters (a, b, and c) indicate significant differences between the initial and final values of the soils based on a least significant difference test at a 5% significance level.

**Figure 3 plants-09-00205-f003:**
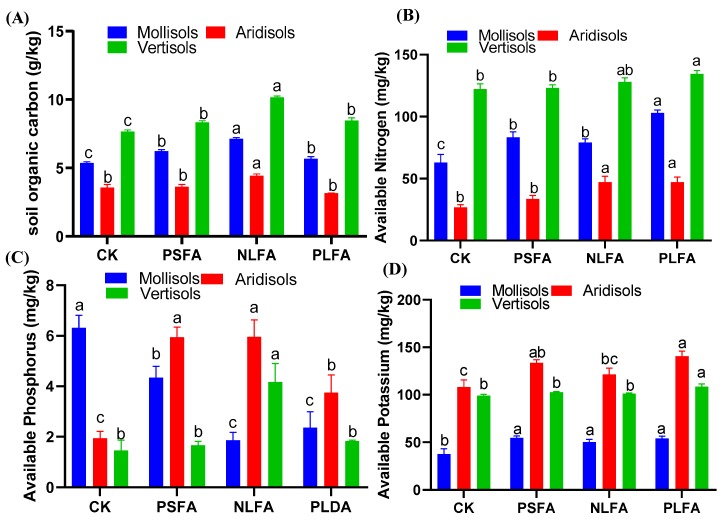
The effect of plant-derived solid fulvic acid (PSFA), mineral-derived liquid fulvic acid (NLFA), and plant-derived liquid fulvic acid (PLFA) on (**A**) soil organic carbon, (**B**) available nitrogen, (**C**) available phosphorus and (**D**) available potassium content of Mollisols, Aridisols, and Vertisols soils. The means ± standard errors are shown (*n* = 4). Different letters (a, b, and c) indicate significant differences between the initial and final values of the soils based on a least significant difference test at a 5% significance level.

**Figure 4 plants-09-00205-f004:**
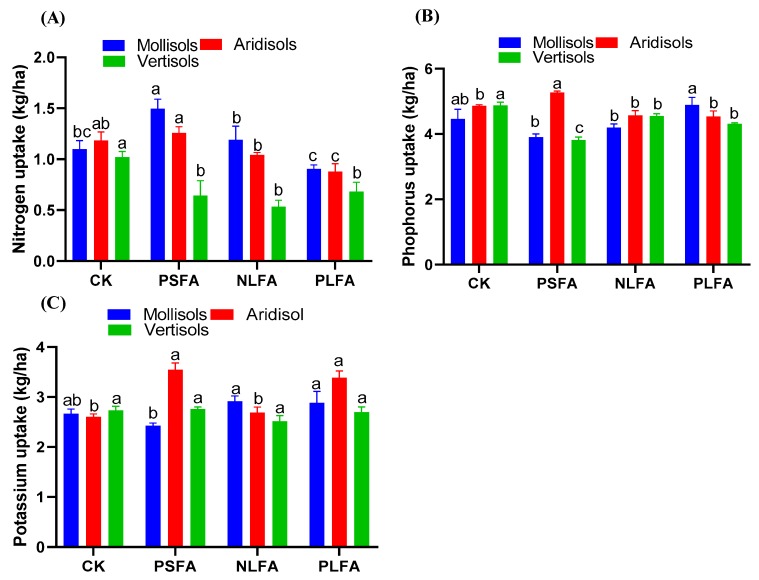
The effect of plant-derived solid fulvic acid (PSFA), mineral-derived liquid fulvic acid (NLFA), and plant-derived liquid fulvic acid (PLFA) on (**A**) nitrogen, (**B**) phosphorus and (**C**) potassium uptake of wheat grown in Mollisols, Aridisols, and Vertisols soils. The means ± standard errors are shown (*n* = 4). Different letters (a, b, and c) indicate significant differences between the initial and final values of the soils based on a least significant difference test at a 5% significance level.

**Figure 5 plants-09-00205-f005:**
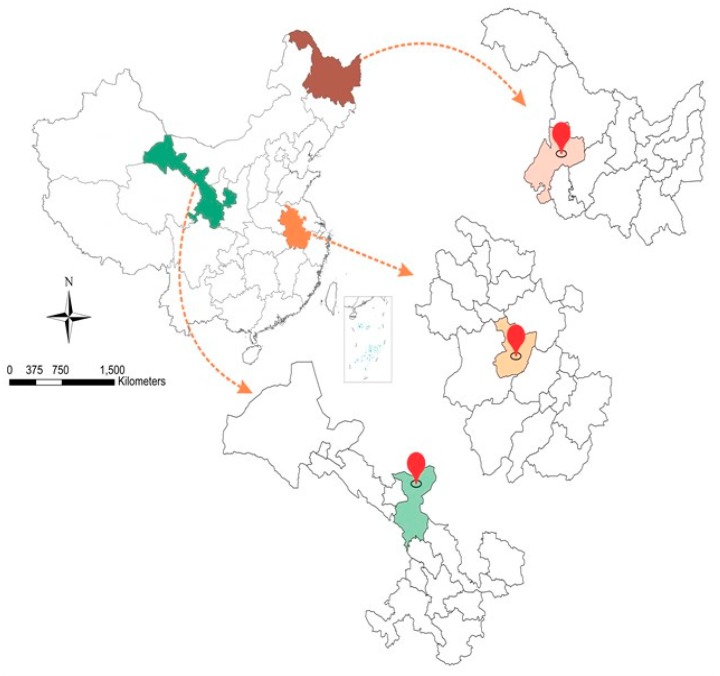
The map showing the purple (Mollisols), yellow (Vertisols), and green (Aridisols) soils of the Heilongjiang, Gansu and Anhui Provinces.

**Table 1 plants-09-00205-t001:** Effect of FAs on organic–inorganic degree compounds and organic–inorganic composites (g/kg).

FA	Oganic–Inorganic Degree Compound (g/kg)			Orgnanic–Inorganic Composite (g/kg)		
	Mollisols	Aridisols	Vertisols	Mollisols	Aridisols	Vertisols
CK	71.85 ± 5.41 ^a^	46.9 ± 1.36 ^c^	53.0 ± 1.39 ^c^	0.30 ± 0.03 ^a^	0.14 ± 0.006 ^b^	0.497 ± 0.03 ^a^
PSFA	44.7 ± 2.42 ^b^	61.3 ± 1.04 ^b^	65.2 ± 1.24 ^b^	0.22 ± 0.008 ^b^	0.22 ± 0.009 ^a^	0.432 ± 0.04 ^ab^
NLFA	49.7 ± 0.65 ^a^	61.7 ± 2.3 ^b^	74.6 ± 1.42 ^a^	0.28 ± 0.01 ^ab^	0.22 ± 0.006 ^a^	0.422 ± 0.02 ^b^
PLFA	61.9 ± 5.3 ^a^	68.8 ± 0.86 ^a^	68.4 ± 1.78 ^b^	0.34 ± 0.02 ^a^	0.21 ± 0.009 ^a^	0.452 ± 0.01 ^ab^

Values are the means ± standard error (*n* = 4) for PSFA (plant-derived solid fulvic acid), NLFA (mineral-derived liquid fulvic acid), and PLFA (plant-derived liquid fulvic acid). Different letters (a, b, and c) indicate significant differences between the soils (*p* < 0.05).

**Table 2 plants-09-00205-t002:** Effect of FA on wheat grain weight and the spike grain weight of wheat (g pot^−1^).

FA	Thousand Grain Weight (g)			Spike Grain Weight (g)		
	Mollisols	Aridisols	Vertisols	Mollisols	Aridisols	Vertisols
CK	49.7 ± 1.9 ^a^	55.79 ± 2.55 ^a^	53.4 ± 0.12 ^a^	99 ± 6.55 ^b^	76.7 ± 5.57 ^ab^	138.4 ± 8.23 ^a^
PSFA	50.5 ± 2.0 ^a^	47.0 ± 1.89 ^b^	47.3 ± 1.12 ^b^	123.2 ± 6.63 ^a^	30.3 ± 5.5 ^b^	124.2 ± 8.85 ^ab^
NLFA	46.5 ± 0.57 ^a^	51.1 ± 1.27 ^ab^	46.2 ± 188 ^b^	97.2 ± 4.63 ^b^	87.1 ± 5.73 ^a^	126.8 ± 5.23 ^ab^
PLFA	47.2 ± 1.21 ^a^	47.2 ± 1.49 ^b^	50.1 ± 1.15 ^a^	110.4 ± 8.20 ^ab^	68.5 ± 9.21 ^ab^	76.7 ± 8.45 ^b^

Values are the means ± standard error (*n* = 4) for PSFA (plant-derived solid fulvic acid), NLFA (mineral-derived liquid fulvic acid), and PLFA (plant-derived liquid fulvic acid). Different letters (a, b, and c) indicate significant differences between the soils (*p* < 0.05).

**Table 3 plants-09-00205-t003:** Effect of FAs on the plant height and plant biomass content of wheat (g pot^-1^).

FA	Plant Height (cm)			Plant Biomass (g)		
	Mollisols	Aridisols	Vertisols	Mollisols	Aridisols	Vertisols
CK	78.2.0 ± 0.91 ^b^	68.9.5 ± 1.26 ^b^	76.7 ± 1.78 ^b^	160.6 ± 1.75 ^d^	153.3 ± 4.37 ^a^	207.4 ± 1.73 ^b^
PSFA	87.0 ± 2.65 ^a^	71.2 ± 1.65 ^b^	84 ± 1.35 ^a^	194.2 ± 4.08 ^b^	82.6 ± 1.42 ^b^	178.3 ± 6.40 ^c^
NLFA	84.2 ± 0.75 ^ab^	88.7 ± 1.11 ^a^	88.2 ± 1.18 ^a^	179.6 ± 3.89 ^c^	166.3 ± 7.29 ^a^	253.0 ± 3.76 ^a^
PLFA	81.6 ± 1.17 ^b^	74.3 ± 1.49 ^b^	82.7 ± 3.03 ^a^	206.3 ± 1.84 ^a^	161.1 ± 9.16 ^a^	210.4 ± 1.8 2 ^b^

Values are the means ± standard error (*n* = 4) for PSFA (plant-derived solid fulvic acid), NLFA (mineral-derived liquid fulvic acid), and PLFA (plant-derived liquid fulvic acid). Different letters (a, b, c, and d) indicate significant differences between the soils (*p* < 0.05).

**Table 4 plants-09-00205-t004:** Properties of the soils used in the experiment.

Soil	EC us/cm	pH	OM g/kg	CEC cmol/kg	AN mg/kg	AP mg/kg	AK mg/kg	TN g/kg	TP g/kg	TK g/kg	Textural Class
Mollisols	30.1	5.2	8.4	21.6	66.0	0.44	78.0	0.70	0.39	20.4	Silty clay
Aridisols	2063	8.4	2.2	4.6	19.0	2.4	525.3	0.20	0.39	20.1	Sandy Loam
Vertisols	132	7.9	11.7	25.2	52.0	1.19	186.6	0.83	0.39	17.3	Clay loam

**Table 5 plants-09-00205-t005:** Elemental composition of plant- and mineral-derived fulvic acid.

FA Type	N	C	H	S
	%			
PSFA	5.39	25.31	5.75	8.47
NLFA	10.29	52.476	9.74	14.84
PLFA	10.78	50.61	11.56	16.96

PS = plant-derived solid, NL = mineral-derived, and PL = plant-derived fulvic acids.

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
