# Peer review of "The Short-Term Effects of Mineral- and Plant-Derived Fulvic Acids on Some Selected Soil Properties: Improvement in the Growth, Yield, and Mineral Nutritional Status of Wheat (Triticum aestivum L.) under Soils of Contrasting Textures"

_plants, 2020, doi:10.3390/plants9020205_

Round 1

Reviewer 1 Report

See WORD file - which I do hope is correctly attached. 'Cut and Paste' did not seem to work? I therefore used the "Choose File" box and that inserted the file name (and hopefully the full file).

Author Response

Response to Comments on Manuscript (NO. plants-701953)

Dear Editors and Reviewers:

Thank you for your letter and for the reviewers’ comments concerning our manuscript entitled “Short-Term Effects of Natural and Plant-Derived Fulvic Acids on Some Selected Soil Properties, Improvement in Growth, Yield and Mineral Nutritional Status of Wheat (Triticum aestivum L.) Under Soils of Contrasting Texture” (No. plants-701953). The comments are all-valuable and are very helpful for revising and improving our paper, as well as guiding our future research. We have reviewed the comments carefully and have made corrections to the manuscript that we hope meet with approval. Revised portions are highlighted in the paper. We have also edited our manuscript through MDPI English Editing services. The main corrections in the paper and the responses to the reviewer’s comments are as follows:

Reviewer #1

General comments

Comment: The manuscript purports to report on the effects of fulvic acids on the properties of three soil types and the growth of wheat in pots filled with such soils. A serious criticism is the lack of adequate, specific information about the sources and analysis of fulvic acids. Organic matter was separated into heavy and light fractions which were analysed for C, H, O and N but the organic acids/compounds were not analysed. Similarly, the Introduction neither describes nor reviews the various organics and their sources. Other researchers could not duplicate this work given this lack of information.

Response: Thank you for pointing it out. We are sorry that there is no description clearly causing confusion. We have added the specific information about the sources and analysis of fulvic acids in the Materials and Methods. In this manuscript, we focused on the effect of FA on carbon in heavy and light fractions, so we did not analyze the H, O and organic acids/compounds. Additionally, We have enriched the content in the Introduction and increased the content of the various organics and their sources.

Comment: The English is poor and the text somewhat wordy. The authors must be commended for striving to write in English and have tried to use a wide vocabulary. Regrettably, some words are inappropriate and/or compromise clarity. The inappropriate use of singular/plural is common. The authors clearly need the kind assistance of a native English speaker from the plant/soil science ranks who is also an experienced science writer.

Response: Thank you for your recognition of our articles. In order to minimize terminology and syntax errors, we have re-invited two native English-speaking professors who specialize in soil and environmental science to help us proofread the revised manuscript.

Comment: I have edited the Introduction and Methods and Materials sections but as there is so much vital information that is lacking or unclear I am unable to progress and believe the manuscript is below the standard expected for submission and review.

Response: Thank you for your kind help. We have modified the full text in accordance with your suggestions, and our English-speaking experts have refined it.

Specific comments

Comment: L43, what is meant by “representing” in this context?

Response: Thank you for referring this question. We have revised the “containing” on behalf of the representing in the reviewed manuscript.

Comment: L44, addicts?

Response: Thanks for your suggestion. We have revise addicts as stops in the reviewed manuscript.

Comment: L45, “restricts” would be the appropriate term- not ‘prevents’

Response: Thank you for referring the grammatic mistakes. We have revised the restricts in the manuscript.

Comment: L49, use ‘these’ – not “this”

Response: Thank you for referring this grammatical mistake. We have revised these in the revised manuscript.

Comment: L50 Delete “the better”

Response: Thank you for referring this mistake. We have deleted the better.

Comment: L50-51 This sentence is quite unclear. Do you mean ‘locally referred to as Shahjiang black and approximately ??? ha are cultivated near…..’ ?.

Response: Thank you for your kind suggestion. We have revised this sentence in our manuscript as locally Vertisols soils are called as Shahjiang black and……….

Comment: L51, delete “Similarly” and start a new paragraph, one devoted to Mollisols. Start the para with ‘Mollisol soils accumulate illite….’

Response: Thank you for your suggestions. We have deleted the similarly and rewrite the sentences as Mollisols soils accumulate illite and vermiculite minerals.

Comment: L53 “alternation”

Response: Thank you for your suggestions. We have changed alternation as retention in our revised manuscript.

Comment: L60 Insert ‘although’ after “Nevertheless”

Response: Thank you for your consideration. We have insert although in our revised manuscript.

Comment: L61 ‘has’ not “have”.

Response: Thank you for point out grammatic mistakes. We have changed has on behalf of have in our manuscript.

Comment: L62 insert ‘such as’ after “practices” and complete the sentence. Delete “becomes imperative” and replace with ‘is necessary’.

Response: Thank you for point out grammatic mistakes. We have changed has on behalf of have in our manuscript.

Comment: L63 Insert ‘the’ before “productive”. Insert after “way”, ‘that sustains agricultural production.

Response: Thank you for suggestions. We have inserted the above-mentioned changing.

Comment: L64 Delete “However”. L65 Delete “Moreover”.

Response: Thank you for suggestions. We have deleted the word However and Moreover in revised manuscript.

Comment: L66 reword: replace with ‘account for half of their total area’.

Response: Thank you for your comments. We have reworded the sentence according to your suggestions.

Comment: L67 Insert ‘soils’ after “productive”. Delete “yet”. Write ‘they’, not “these”. Insert ‘the’ after “are”. Delete “Moreover” and “long-term sustainability scarcer”.

Response: Thank you for your comments. We have inserted the soils and deleted yet and wrote they and deleted Moreover in the manuscript.

Comment: L68 Rewrite: ‘Water availability, salinization and wind erosion threaten the productive capacity of Aridisols for agriculture [14, 15, 16]. They are vulnerable to erosion which depletes top soil, lowering the content of clay and organic matter and the availability of …….’.

Response: Thank you for your comments. We have re-write the sentence according to your suggestions in the revise manuscript.

Comment: L72 ‘A significant number’ rather than “amount”. Delete “Hitherto”.

Response: Thank you for your comments. We have modified the manuscript and delete the Hitherto.

Comment: L73 Insert ‘the’ before “above”. L77 Insert ‘of the’ after “constituent”. L79 use ‘most’ rather than “furthermost”.

Response: Thank you for your suggestions. We have Inserted the required word as you commented in our manuscript.

Comment: L83 What do you mean by “divided” in this context? Do you mean ‘more understood when they are considered separately as FA, HA and humin’?

Response: Thank you for pointed out. According to this context we mean that “these substances become more effective when they were separated in the forms of fulvic acid. Humic acids and humin.

Comment: L84 What do “represent” and “deposits” mean in this sentence?¸ Write ‘can’ rather than “have been seen to”.

Response: Thank you for comments. This sentence was not giving any meaning here so we deleted and also rewrote can as you mentioned in the revised manuscript.

Comment: L88 “researcher” - should be plural. But plural is not appropriate given that you cite only one study.

Response: Thank you for comments. We have changed the word in the plural forms.

Comment: L89 use ‘increases’ not “rouses” Insert ‘and’ before “cell”.

Response: Thank you for suggestion. We have used Increases in the revised manuscript.

Comment: L91 Reword: ‘the positive effects of applying organic amendments to soils used for agriculture will be influenced by the……’

Response: Thank you for suggestion. We have re-write the sentences as you mentioned in the revised manuscript.

Comment: L92 Delete “added”. Insert ‘they contain’. L93 Use ‘vary’ rather than “cause”.

L96 Rather than “to find out”, write ‘(1) measure’. L97 Delete “the”. Insert ‘and’ after “properties”.

Response: Thank you for suggestion. We have modified all these comments in the revised manuscript.

Comment: L98 parameter should be plural. Write ‘mineral content’ rather than “minerals elements”. Reword last sentence, eg. ‘(2) To examine the interaction of fulvic acids and soil types’. BUT, the types of FA are not described and reviewed??

Response: Thank you for suggestion. We have changed the word parameter as plural, mineral content and reword the sentences according to your suggestion in the revised manuscript.

Comment: L367 Delete last two words and the following two on L368. Cite a reference after “methods” and delete rest of sentence. Sentences 2 and 3 of Section 4.2 are also unclear and need to be rewritten.

Response: Thank you for suggestion. We have re-write the sections 4.2. and changed the sentence 2 and 3 in the revised manuscript.

Comment: Poor English: eg. “Shacked, rmp, mints” L368 “considered” is not the right word.

Response: Thanks for your suggestion. We are sorry there was some typing errors, we have revised the words in the manuscript.

Comment: L391-3 This sentence is quite unclear.

Response: Thanks for your suggestion. We are sorry there was some typing errors, we have revised the words in the manuscript.

Comment: L394 What rate of fertilizer was applied? Express nutrients as elements, not as compounds.

Response: Thank your pointed out this mistake. We have inserted the applied rate of Fertilizer in the revised manuscript.

Comment: L396-8 The description of FAs is vague and inadequate. Cannot fulvic acids be analyzed using infrared spectrometry?

Response: Thank your comments. We never mentioned that Fulvic acid was analyzed by infared spectrometry in our manuscript but still want to know your question.

Comment: L 401 Insert references.

Response: Thanks for your suggestion. We have inserted the reference in the revised manuscript.

Comment: L403 what does “those related to” mean in this context?

Response: Thanks for your comments. We have deleted the word those related and revised the sentence in the revised manuscript.

Comment: L413 Surely you mean you analysed the replicate data, not “means”.

Response: Thank for point out. No as I mentioned in the manuscript we analysed means of the replicate in the manuscript.

Reviewer #2

Comment: The study “Short-Term Effects of Natural and Plant-Derived Fulvic Acids on Some Selected Soil Properties, Improvement in Growth, Yield and Mineral Nutritional Status of Wheat (Triticum aestivum L.) Under Soils of Contrasting Texture” deals with wheat grown on three soils having silty clay, sandy loam and clay loam texture, the soils were treated with FAs derived from plant and natural materials. The topic of the manuscript is interesting and the authors had a good idea for a research project. The subject is relevant, the analytical methodologies are adequate and the volume of data seems to be enough for publication. Methodology is well explained. I have no hesitation in recommending publication following minor revision.

Response: Thank you for your careful reading of our manuscript and affirming our research work. In order to make the manuscript better, we have revised the manuscript and invited experts who are native English speakers to edit the new manuscript.

We tried our best to improve the manuscript through careful revisions and help from a scientific editor who is a native English speaker. These changes do not influence the content and framework of the paper. In this response letter, we did not list the specific changes but instead highlighted them in the revised paper.

We appreciate the Editors/Reviewers’ work, and hope that the corrected manuscript will meet with approval.

Once again, thank you very much for your comments and suggestions.

Sincerely yours,

Prof. Xibai Zeng

Reviewer 2 Report

The study “Short-Term Effects of Natural and Plant-Derived Fulvic Acids on Some Selected Soil Properties, Improvement in Growth, Yield and Mineral Nutritional Status of Wheat (Triticum aestivum L.) Under Soils of Contrasting Texture” deals with wheat grown on three soils having silty clay, sandy loam and clay loam texture, the soils were treated with FAs derived from plant and natural materials. The topic of the manuscript is interesting and the authors had a good idea for a research project. The subject is relevant, the analytical methodologies are adequate and the volume of data seems to be enough for publication. Methodology is well explained. I have no hesitation in recommending publication following minor revision.

General comments:    

-Introduction:

The introduction is well conducted since it addresses the issue and refers to relevant literature.

-Material and methods:

The methodology is well thought through.

- Results and Discussion:

In my point of view this section is drawn up well. Comparison of the results with other authors’ conclusions would be appreciated.

Please unify the use of abbreviations.

- Conclusions are too concise.

The paper should be published with minor revisions in order to reach the standard quality for publication.

Author Response

Response to Comments on Manuscript (NO. plants-701953)

Dear Editors and Reviewers:

Thank you for your letter and for the reviewers’ comments concerning our manuscript entitled “Short-Term Effects of Natural and Plant-Derived Fulvic Acids on Some Selected Soil Properties, Improvement in Growth, Yield and Mineral Nutritional Status of Wheat (Triticum aestivum L.) Under Soils of Contrasting Texture” (No. plants-701953). The comments are all-valuable and are very helpful for revising and improving our paper, as well as guiding our future research. We have reviewed the comments carefully and have made corrections to the manuscript that we hope meet with approval. Revised portions are highlighted in the paper. We have also edited our manuscript through MDPI English Editing services. The main corrections in the paper and the responses to the reviewer’s comments are as follows:

Reviewer #2

Comment: The study “Short-Term Effects of Natural and Plant-Derived Fulvic Acids on Some Selected Soil Properties, Improvement in Growth, Yield and Mineral Nutritional Status of Wheat (Triticum aestivum L.) Under Soils of Contrasting Texture” deals with wheat grown on three soils having silty clay, sandy loam and clay loam texture, the soils were treated with FAs derived from plant and natural materials. The topic of the manuscript is interesting and the authors had a good idea for a research project. The subject is relevant, the analytical methodologies are adequate and the volume of data seems to be enough for publication. Methodology is well explained. I have no hesitation in recommending publication following minor revision.

Response: Thank you for your careful reading of our manuscript and affirming our research work. In order to make the manuscript better, we have revised the manuscript and invited experts who are native English speakers to edit the new manuscript.

General comments:    

Comment:

Introduction: The introduction is well conducted since it addresses the issue and refers to relevant literature.

Material and methods: The methodology is well thought through.

Results and Discussion: In my point of view this section is drawn up well. Comparison of the results with other authors’ conclusions would be appreciated.

Response: Thank you for your careful reading of our manuscript and acknowledging our research and work

Comment: Please unify the use of abbreviations.

Response: Thank you for pointing the problem. We have unified the use of abbreviations in the new manuscript.

Comment: Conclusions are too concise.

Response: Thank you for your constructive suggestions. We have modified the conclusions to make it more accurate and scientific.

Comment: The paper should be published with minor revisions in order to reach the standard quality for publication.

Response: Thank you for your recognition and recognition of our manuscript. We have revised the manuscript and invited experts who are native English speakers to edit the new manuscript.

We tried our best to improve the manuscript through careful revisions and help from a scientific editor who is a native English speaker. These changes do not influence the content and framework of the paper. In this response letter, we did not list the specific changes but instead highlighted them in the revised paper.

We appreciate the Editors/Reviewers’ work, and hope that the corrected manuscript will meet with approval.

Once again, thank you very much for your comments and suggestions.

Sincerely yours,

Prof. Xibai Zeng

Round 2

Reviewer 1 Report

GENERAL

The short turnaround time did not deliver a major revision which was suggested.  Not surprisingly, you have missed or overlooked some points offered previously.

I am a foreign research agronomist who has no direct experience with humus/fulvic acid fertilizer. Such a demographic is probably an important sector of the readership of the Journal, PLANTS. Please help such readers by providing a clear, detailed M&M.

I have a keen interest to understand this - to me, novel - paper and need to understand what you did and how you did it. I read the abstract and Introduction and then the Methods and Materials before looking at Results. My struggle to understand the Materials and Methods continues.  English still needs checking.

ABSTRACT

Terms:  Plants are part of nature. I struggle with what you seek to distinguish  between “plant” and “natural”. I strongly recommend you drop all reference to “natural” and replace it with ‘mineral’. I note that the first author has in fact already made this change in his 2019 paper that was published in MOLECULES 24:1535.

Quantification:  Apart from increases in C, none of the results you describe are qualified by quantity.  You simply say “increased” or “increases” re available N, P and K. The potential reader will not be excited to read the full text by this lack of detail.  You need to arouse interest in agricultural significance.  Can you therefore insert ‘increased by 10%’ or whatever it was? Or, ‘available P was increased from 2 to 6 mg/kg (P<0.05) depending on FA source and soil type etc.

English: Potential readers may be put off by the poor expression in the opening two sentences. Reword carefully.

After “showed that” I would insert “in treated soils” and then, after “60%”, delete the remainder of that sentence. Write ‘increased in treated soil, and P….’

What do you mean by “organic-inorganic”?  Do you mean organic and inorganic compounds/composites?

Lines 33-37.  Change to past tense. 

INTRODUCTION

L68 ‘their’ total area not “the total area”

L84 Do you mean that they can be better understood (rather than be more effective) when the various components are considered separately?

L90-91 None of these references specifically mention fulvic acids in their titles; they refer to OM or more general components.

L101-2  What types of FA? So, what types Have been described and reviewed?

Is it accurate to say “these types of fulvic acids have not been described or reviewed”?

MATERIALS AND METHODS

It is helpful to describe the design of the experiment and the nature and source of the soils and treatment material before describing the methods of pot management and measurements.

The 1st paragraph is about the soils – as the sub-heading indicates, except that on L353 three sentences appear that concern leaves? That does not fit.

L364 delete second “soil”.

Section 4.2:  What is this soil OM? Please clarify. Is it from pre- or post-treatment?

L390 What do you mean by “alternated”?

L392  As mentioned in the first review, the nutrients should be described as elements not oxides. That is the common International convention.

L394  Replace “from two” with ‘both a’

L396 Please can you insert a (2nd) sentence that describes the FA used in some detail. The Shandong Co. would surely have a detailed analysis of their fertilizer product. How was it extracted? What is its purity in the commercial product you used?

L398-9  Also, briefly outline the method - before the citation.

RESULTS

Fig 1B data difficult to appraise due to the slight differences.  Better presented as a Table.

What are "degree compounds"?

DISCUSSION

L238  Objectivity:  Delete "It has been proven by many scientists that". Just make the statement and support it with references. 

Generally the English in this section is good but still needs careful checking in places, e.g. L239-40 - plants are absorbed??

CONCLUSION

This section should only be about the research you have reported in this paper. Remove all statements about FA that were not assessed in this study (sentence 2 for example).  Such points can be stated (with references) in the introduction/discussion - but not in the conclusion. 

The increased uptakes of N, P and K are not great.  So, in the last sentence, after “in the field” can you add ‘and compared for cost-effectiveness with alternative fertilizers’.  Without such comparison, the need to research the mechanisms does not appear to be urgent.

L428 re. “properties of A. and V. in soils”.  I would write as ‘properties for A. and V. soils’.

Author Response

Response to Comments on Manuscript Number: (plants-701953)

Title: The Short-Term Effects of Natural and Plant-Derived Fulvic Acids on Some Selected Soil Properties: Improvement in the Growth, Yield, and Mineral Nutritional Status of Wheat (Triticum aestivum L.) Under Soils of Contrasting Textures

Dear Editor,

We are thankful to the editor and the anonymous reviewer for providing the valuable comments on our manuscript. We have thoroughly revised our manuscript and have tried our best to incorporate all the changes suggested by the respected reviewers in a scientifically sound manner. All suggestions have been addressed (please see the highlighted text) in the revised manuscript. Authors detailed point-by-point response to the editor and reviewer’s comments are given below. Moreover, in order to improve the overall quality, significant additions/deletions and modifications have been carried out in the revised version.

We would like to thank you in advance for the effort done to review the manuscript which was very helpful for us to improve its quality.

Yours Sincerely,

Authors

General comments

Comment: The short turnaround time did not deliver a major revision which was suggested. Not surprisingly, you have missed or overlooked some points offered previously.

Response: We are very sorry that our revised manuscript can’t satisfy you! We have revised the manuscript again according to your comments, in order to better meet the requirements of the journal.

 Comment: I am a foreign research agronomist who has no direct experience with humus/fulvic acid fertilizer. Such a demographic is probably an important sector of the readership of the Journal, PLANTS. Please help such readers by providing a clear, detailed M&M.

I have a keen interest to understand this - to me, novel - paper and need to understand what you did and how you did it. I read the abstract and Introduction and then the Methods and Materials before looking at Results. My struggle to understand the Materials and Methods continues. English still needs checking.

Response: We appreciate your time and efforts devoted to review the manuscript and suggesting very helpful comments. The details on FA’s sources, chemical composition and related concerns are now referred in the revised version M&M section. However, for clarity and further reading our previous paper have been referred in M&M section. Moreover, English improvement have been further carried out throughout the MS.

Comment: Plants are part of nature. I struggle with what you seek to distinguish  between “plant” and “natural”. I strongly recommend you drop all reference to “natural” and replace it with ‘mineral’. I note that the first author has in fact already made this change in his 2019 paper that was published in MOLECULES 24:1535.

Response: We agree with the reviewer. Thank you for pointing it out. We have replaced the “natural” with the “mineral” in the revised manuscript.

Comment: Quantification: Apart from increases in C, none of the results you describe are qualified by quantity.  You simply say “increased” or “increases” re available N, P and K. The potential reader will not be excited to read the full text by this lack of detail.  You need to arouse interest in agricultural significance.  Can you therefore insert ‘increased by 10%’ or whatever it was? Or, ‘available P was increased from 2 to 6 mg/kg (P<0.05) depending on FA source and soil type etc.

Response: Yes, we agree with the reviewer. Thank you for the nice suggestion. We have quantitatively described the effects of fulvic acid on soil nutrients. The required improvements are now addressed in the revised version. Please see highlighted track changed text throughout the MS taxt.

 Comment: English: Potential readers may be put off by the poor expression in the opening two sentences. Reword carefully.

Response: Thank you for your comments. We have modified these sentences for a clear and broad readership.

 Comment: After “showed that” I would insert “in treated soils” and then, after “60%”, delete the remainder of that sentence. Write ‘increased in treated soil, and P….’

Response: Thank you for your nice suggestions. Required corrections are now addressed.

Comment: What do you mean by “organic-inorganic”?  Do you mean organic and inorganic compounds/composites?

Response: Thank you for your consideration. It is clearly written “organic–inorganic composites decreased in Vertisols relative to the other two soils” in re revised manuscript.

 Comment: Lines 33-37.  Change to past tense.

Response: Thank you for your suggestions. We have modified this sentence in past tense.

 Comment: L68 ‘their’ total area not “the total area”

Response: Thank you for your comments. We have modified as their total area in the revised manuscript.

 Comment: L84 Do you mean that they can be better understood (rather than be more effective) when the various components are considered separately?

Response: Thank you for your consideration. The sentence has now been revised in the final version.

Comment: L90-91 None of these references specifically mention fulvic acids in their titles; they refer to OM or more general components.

Response: Thank you for your comments. Dear reviewer, the references are correctly cited. As, OM is the main source of Hs in the soil. We have replaced the word fulvic acid with humic substances to better track for the required information provided.

 Comment: L101-2 What types of FA? So, what types Have been described and reviewed? Is it accurate to say “these types of fulvic acids have not been described or reviewed”?

Response: Thank you for your comments. We have modified the sentence in the revised manuscript.

 Comment: It is helpful to describe the design of the experiment and the nature and source of the soils and treatment material before describing the methods of pot management and measurements.

Response: Thank you for your suggestion. We have already described the experimental design and nature source of the soils and treatments in the M&M sections.

 Comment: The 1st paragraph is about the soils – as the sub-heading indicates, except that on L353 three sentences appear that concern leaves? That does not fit.

Response: Thank you for comments. We have modified the 1st paragraph as collection and preparation of soil and plant samples in the revised manuscript.

Comment: L364 delete second “soil”.

Response: Thank you for your comments. We have modified and deleted the second soil the revised manuscript.

Comment: Section 4.2: What is this soil OM? Please clarify. Is it from pre- or post-treatment?

Response: Thank you for your comments. We have analyzed the post treatments for Soil organic matter.

Comment: L390 What do you mean by “alternated”?

Response: Thank you. It was a typo. Now corrected.

Comment: L392 As mentioned in the first review, the nutrients should be described as elements not oxides. That is the common International convention.

Response: Thank you for your comments. The Compound Fertilizer was provided by well reputed company and they have the composition of Nutrients in the form of P2O5 and K2O.

 Comment: L394 Replace “from two” with ‘both a’

Response: Thank you for your comments. We have modified and replace “from two” with ‘both’ in the revised manuscript.

Comment: L396 Please can you insert a (2nd) sentence that describes the FA used in some detail. The Shandong Co. would surely have a detailed analysis of their fertilizer product. How was it extracted? What is its purity in the commercial product you used?

Response: Thank you for your suggestions. We have tired to put some information related FA and Shandong Quan company products in the revised manuscript. Moreover, we have also referred our previous study regarding the detailed information of the FAs used in present study.

 Comment: L398-9 Also, briefly outline the method - before the citation.

Response: Thank you for your consideration. The suggestions are now incorporated. .

Comment: Fig 1B data difficult to appraise due to the slight differences. Better presented as a Table.

Response: Thank you for your suggestion. We would like to go with the figure perhaps as more easy to differentiate between different soils treated with various FAs irrespective of the marked differences.

Comment: What are "degree compounds"?

Response: Thank you for your point. We have removed the word degree.

 Comment: L238 Objectivity:  Delete "It has been proven by many scientists that". Just make the statement and support it with references.

Response: Thank you. We have revised the whole sentence.

Comment: Generally, the English in this section is good but still needs careful checking in places, e.g. L239-40 - plants are absorbed??

Response: Thank you for your comments. We have modified this part further.

Comment: This section should only be about the research you have reported in this paper. Remove all statements about FA that were not assessed in this study (sentence 2 for example). Such points can be stated (with references) in the introduction/discussion - but not in the conclusion.

Response: Thank you for you comments. We have modified and deleted all the unnecessary information related FAs in this section in the revised manuscript.

Comment: The increased uptakes of N, P and K are not great. So, in the last sentence, after “in the field” can you add ‘and compared for cost-effectiveness with alternative fertilizers’. Without such comparison, the need to research the mechanisms does not appear to be urgent.

Response: Thank you for your suggestion. The objective of our study was not to evaluate the cost-effectiveness of applied treatments relative to other fertilization practices. Therefore, we have not calculated cost-effectiveness analysis at the moment, but have provided a theoretical basis of understanding responses of FAs on varying soils that can support crop production.

Comment: L428 re. “properties of A. and V. in soils”.  I would write as ‘properties for A. and V. soils’.

Response: Thank you so much for you comments. We have modified as properties for Aridisols and Vertisols soils in the revised manuscript.